# Comparative Effects of Heat Stress at Booting and Grain-Filling Stage on Yield and Grain Quality of High-Quality Hybrid Rice

**DOI:** 10.3390/foods12224093

**Published:** 2023-11-11

**Authors:** Xinzhen Zhang, Qiuping Zhang, Juan Yang, Yuhao Jin, Jinshui Wu, Hang Xu, Yang Xiao, Yusha Lai, Zhiqiang Guo, Jianlong Wang, Wanju Shi

**Affiliations:** College of Agronomy, Hunan Agricultural University, Changsha 410128, China; zxinzhen@outlook.com (X.Z.); zhangqiuping@hunau.edu.cn (Q.Z.); yjhunau@163.com (J.Y.); a336688550526@163.com (Y.J.); wu18890127172@163.com (J.W.); xuhdyx@163.com (H.X.); 15576173207@163.com (Y.X.); lys2524683378@163.com (Y.L.); gzq256410@126.com (Z.G.); wjl9678@126.com (J.W.)

**Keywords:** booting stage, grain-filling stage, grain qualities, higher temperature, yield

## Abstract

Rice plants are highly sensitive to high-temperature stress, posing challenges to grain yield and quality. However, the impact of high temperatures on the quality of high-quality hybrid rice during the booting stage, as well as the differing effects of the booting and grain-filling stages on grain quality, are currently not well-known. Therefore, four high-quality hybrid rice were subjected to control (CK) and high-temperature stress during the booting (HT1) and grain-filling stages (HT2). Compared to the control, HT1 significantly reduced the spikelets panicle^−1^ (16.1%), seed setting rate (67.5%), and grain weight (7.4%), while HT2 significantly reduced the seed setting rate (6.0%) and grain weight (7.4%). In terms of quality, both HT1 and HT2 significantly increased chalkiness, chalky grain rate, gelatinization temperature, peak viscosity (PV), trough viscosity (TV), final viscosity (FV), and protein content in most varieties, and significantly decreased grain length, grain width, total starch content, and amylose content. However, a comparison between HT1 and HT2 revealed that the increase in chalkiness, chalky grain rate, PV, TV, and FV was greater under HT2. HT1 resulted in a greater decrease in grain length, grain width, total starch content, and amylose content, as well as an increase in protein content. Additionally, HT1 led to a significant decrease in amylopectin content, which was not observed under HT2. Therefore, future efforts in breeding and cultivating high-quality hybrid rice should carefully account for the effects of high temperatures at different stages on both yield and quality.

## 1. Introduction

In light of the expanding global population and rising food consumption [1], rice, a major staple crop worldwide, is projected to experience a demand surge of 35% to 56% by 2050 [2]. Developing hybrid rice varieties has proven effective in meeting this demand, providing clear yield advantages over conventional rice and significantly improving rice production [3]. The annual cultivation area of hybrid rice has increased rapidly, accounting for over half of the total rice cultivated area in China [4], and is extended to other countries such as Vietnam, India, Bangladesh, and the Philippines [5]. However, early-bred hybrid rice varieties faced challenges in terms of quality and taste compared to conventional rice, hindering their development [6]. The increased demand for high-quality hybrid rice due to rising living standards has led to the production of superior-quality hybrid rice varieties [7]. In the fields, however, hybrid rice yield and grain quality are sensitive to changes in the climates, particularly sudden extreme high-temperature events [8,9,10]. High temperatures during the reproductive stage in four planting areas of South China between 1981 and 2010 reduced rice yield by 1.5–9.7% [11]. Additionally, a 1 °C increase in average temperature during the rice growing season resulted in a 6.2% paddy yield reduction and an 8.1–11.0% decline in total milling revenue by 8.1–11.0% [12]. Hybrid rice exhibits a greater vulnerability to elevated temperature stress [13,14], increasing concerns about its widespread adoption, especially given the rising frequency of high temperatures due to global climate warming [15]. Therefore, an understanding of the yield and grain quality responses of newly bred high-quality hybrid rice to higher temperature stress is essential to ensure food security worldwide.

Grain filling represents the final stage of rice growth, during which the fertilized ovaries develop into caryopses. This crucial process is dependent on the transportation and accumulation of storage substances such as starch, storage protein, and lipids within the rice grain, which is highly sensitive to temperatures. Recent studies have explored the response of rice yield and quality to high-temperature stress during the grain-filling stage. These investigations have revealed that high temperatures have a significant impact on the nutritional quality of grains. For instance, there is a decrease in starch content and an increase in protein content under high-temperature conditions [16]. It has been suggested that the reduction in starch assimilates due to high temperatures leads to a decrease in grain length and width [17], and alterations in starch structure and arrangement contribute to increased chalkiness [18]. Additionally, elevated temperatures during the grain-filling stage adversely affect the cooking and eating quality of rice. The reduced starch content caused by high-temperature stress can result in higher gelatinization properties, including gelatinization temperature and ΔH, as well as alterations in pasting viscosities [19,20].

However, the climate is undergoing unprecedented changes globally, with high temperatures becoming increasingly frequent in recent decades. The various stages of rice growth are increasingly subjected to high temperatures [21], for example, the booting stage. High-temperature stress during the booting stage has been shown to reduce grain yield by decreasing grain weight, seed setting rate, and spikelets panicle^−1^ [22]. While previous research conducted by Zhen et al. [23] has revealed the negative impact of high-temperature stress on the amylose content, peak viscosity, and breakdown value of japonica rice, as well as the increase in protein content and gelatinization temperature, limited studies have explored the quality response of high-quality hybrid rice genotypes to high-temperature stress during the booting stage. Notably, hybrid rice has been found to be relatively more susceptible to high temperatures compared to other genotypes [24]. Therefore, it is crucial to investigate the quality changes in high-quality hybrid rice under high-temperature stress during the booting stage to evaluate the harm inflicted on its quality. Additionally, there is currently limited information comparing the differences in yield and quality response of rice under high-temperature stress during both the booting and grain-filling growth stages. Conducting a comparative analysis of heat stress responses in multiple reproductive stages can help establish appropriate sowing and cultivation measures for hybrid rice and effectively cope with sudden high-temperature weather events resulting from climate change.

In this study, four high-quality indica hybrid rice varieties were exposed to control temperature and heat stress during the booting and grain-filling stages. The study aims to analyze the similarities and differences in the response of yield components, appearance quality, thermal properties, pasting properties, and nutritional quality of high-quality hybrid rice to heat stress during the booting and grain-filling stages.

## 2. Materials and Methods

### 2.1. Plant Materials and Experimental Design

The experiment was conducted at the Rice Research Institute of Hunan Agricultural University (14°11′ N, 121°15′ E, 21 m above sea level (asl)), Hunan Province, China. Four high-quality indica hybrid rice varieties, that is, Yueliangyou2646 (YLY2646), Quanyousimiao (QYSM), Yliangyou-2 (YLY2H), and Longliangyouhuanglizhan (LLYHLZ) were selected. The germinated seeds were raised in seedling trays. After twenty-two days, the rice seedlings were transplanted into pots (about 32 cm in height and about 28 cm in diameter) containing 15 kg of paddy soil from 0 to 20 cm of the realistic paddy fields. The soil had a clay loam texture with a pH of 5.68, an organic matter content of 21.54 g kg^−1^, a total N content of 1.87 g kg^−1^, an Olsen *p* value of 47.65 mg kg^−1^, and an available K content of 158.38 mg kg^−1^. In each pot, 5 g of compound fertilizer (N:P_2_O_5_:K_2_O = 25:10:15) was mixed with soil before transplanting, and 2 g of fertilizer was top-dressed at the mid-tillering stage. During growth periods of plants, diseases, insects, and weeds were effectively controlled to plants to prevent plant damage.

At the booting stage (HT1 specifically indicated the 7th and 8th stage of the young panicle differentiation stage in this study) and grain-filling stage (HT2, 10 days after heading), the main tillers of 50 pots of each variety were labeled, and then transferred in temperature-controlled chambers (Zhizhong Company, Beijing, China) for six days of heat stress, while another set of plants were moved into chambers with control temperature. During temperature treatments, the daytime (7:00–19:00) and nighttime (19:00–7:00) temperatures for higher temperature stress (including both HT1 and HT2) were 38 °C (actual 38.2 ± 1.3 °C) and 30 °C (actual 29.7 ± 1.0 °C) and the daytime and nighttime temperatures during control treatment were 30 °C (actual 30.0 ± 1.6 °C) and 22 °C (actual 22.3 ± 1.2 °C). The relative humidity in all chambers was maintained at 75% (actual 73.4 ± 5.2%). A HOBO sensor (HOBO, MX2301, Onset Computer Corporation, Bourne, MA, USA) was placed in the climate chamber to record the air temperature and relative humidity during the temperature treatment period. After temperature treatments, all plants were moved out of the climate chamber and continued to grow in a controlled environment until harvested at maturity.

### 2.2. Sampling and Yield Observations

At maturity, panicles tagged on each individual plant for every variety and temperature treatment were collected to measure the number of spikelets per panicle. Subsequently, the filled and empty spikelets were manually counted in order to determine the seed-setting rate, which represents the ratio of filled grains to the total number of grains. Additionally, six sets of 30 filled grains were subjected to oven-drying until a constant weight was achieved, enabling the measurement of grain weight. Simultaneously, a larger quantity of harvested rice grains was air-dried and stored at room temperature for a duration of three months. Subsequently, the grains were milled to examine rice quality and assess the physicochemical properties of the starch.

### 2.3. Appearance Quality

Three sets of samples, each comprising more than 200 milled rice grains, were randomly selected for scanning using a flatbed scanner (ScanMaker i800plus, MICROTEK, Shanghai, China). The resulting images were then processed utilizing the Wanshen SC-E Rice Appearance Quality Analyzer (Wanshen Detection Technology Co., Ltd., Hangzhou, China). This analysis aimed to measure various parameters, including grain length, grain width, length/width ratio, chalkiness degree (representing the percentage of all chalky grain area in relation to the total projected grain area), and chalky grain rate (representing the percentage of all chalky grains).

### 2.4. Thermal Properties Analysis

The thermal properties of grain starch were determined using a differential scanning calorimetry analyzer (DSC 25, TA Instruments, New Castle, DE, USA) according to the standard ASTM (D-3418). Five milligrams of starch were dispersed with 10 ul of sterile water, and then the mixture was hermetically sealed and left to stand at room temperature for 24 h before being heated in the DSC. The DSC analyzer was calibrated using an empty aluminum pan as a reference. The sample pans were heated at a rate of 10 °C/min from 30 °C to 95 °C. Onset temperature (To), peak temperature (Tp), conclusion temperature (Tc), and gelatinization enthalpy (∆H) were calculated by the TA Universal Analysis 2000 software. The gelatinization temperature range (R) and peak height index (PHI) were calculated as R = Tc − To and PHI = ΔH/(Tp − To), respectively [25].

### 2.5. Determination of Pasting Characteristics

The pasting properties of grain starch were assessed using a Rapid Visco-Analyzer (RVA-3D, Newport Scientific, Warriewood, Australia) in accordance with the national standard of China GB/T24852-2010 [26], issued by the National Standardization Administration of China (https://openstd.samr.gov.cn/bzgk/gb/newGbInfo?hcno=006DFF61495A2606D6AF3360C85B9A6A, accessed on 16 June 2023). The analysis began by accurately weighing rice flour, which was then placed into an RVA sample canister. Ultrapure water was added to the canister along with the rice flour. The canister containing the sample was subsequently transferred to the RVA for testing. During the testing process, the temperature inside the RVA tank followed a specific heating-cooling program. It initially started at 50 °C for 1 min and gradually increased to 95 °C at a rate of 12 °C per minute. Subsequently, the temperature was maintained at 95 °C for 2.5 min before being reduced to 50 °C at a rate of 12 °C per minute. Finally, the temperature was kept constant at 50 °C for 2 min. To analyze the pasting properties, such as peak viscosity (PV), trough viscosity (TV), final viscosity (FV), breakdown value (PV–TV), and setback value (FV–PV), the TCW (Thermal Cline for Windows) program was employed.

### 2.6. Starch and Protein Measurement

The determination of the total starch content was carried out using the total starch kit (Suzhou Comin Biotechnology Co., Ltd., Suzhou, China) following the protocol provided by the kit. The amylose content was measured using the amylose–iodine reaction, referring to the national standards of the People’s Republic of China (GB/T 17891-2017) [27]. For the amylose content measurement, 100 mg of rice flour was mixed with 1 mL of 95% ethanol and 9 mL of 1 M NaOH, and the mixture was then boiled for 10 min. After cooling, the volume was adjusted to 100 mL with distilled water. To 5 mL of the solution, 1 mL of 1 M aqueous acetic acid and 2 mL of iodine solution (0.2 g iodine and 2.0 g potassium iodide in 100 mL aqueous solution) were added. The volume was then adjusted to 100 mL with distilled water, and the absorbance of the solution was measured at 620 nm using a spectrophotometer. The amylopectin content was calculated by subtracting the amylose content from the total starch content. The determination of the total protein content was performed following the method by Lan et al. [28] with slight modifications. In brief, the protein content was indirectly measured using the semi-micro Kjeldahl method to estimate the nitrogen concentration, and a Kjeldahl conversion coefficient of 5.95 was applied, referring to the national standards of the People’s Republic of China (GB/T 5009.5-2016) [29].

### 2.7. Statistical Analysis

The statistical analysis employed in this study involved factorial analysis of variance (ANOVA), which was performed using the R language (version 4.1.0; http://www.R-project.org, accessed on 16 June 2023). Mean values ± standard error were reported in all tables and figures. To determine significantly different values within the same variety, the LSD test with a significance level of *p* < 0.05 was employed. Furthermore, principal component analysis (PCA) and Pearson’s correlation analysis were carried out using the R language. OriginPro2022 (Origin Lab Corporation, Northampton, MA, USA) was utilized for graphical representation.

## 3. Results

### 3.1. Effects of Heat Stress on Yield Parameters

Noticeable differences were observed among varieties, temperature treatments, and their interactions for spikelets panicle^−1^, seed setting rate, and grain weight of brown rice (Figure 1). When compared to the control (CK), the higher temperature during the booting stage (HT1) caused a significant reduction of 16.1%, 67.5%, and 7.4% in spikelets per panicle, seed setting rate, and grain weight, respectively, across the four varieties. Among the selected varieties, YLY2646 exhibited the lowest reductions (12.0%, 51.7%, and 2.0% in spikelets per panicle, seed setting rate, and grain weight, respectively) under the HT1 treatment, while LLYHLZ had the largest reductions of 20.7%, 75.4%, and 13.3% in the same parameters. The higher temperature during the grain-filling stage (HT2 treatment) resulted in a significant average reduction of 6.0% in seed setting rate and 7.4% in grain weight. However, there was no significant effect on spikelets per panicle across the four varieties under the HT2 treatment (Figure 1). Variety YLY2646 experienced the largest decrease in seed setting rate (12.7%) compared to the other varieties when exposed to HT2, while the most significant reduction in grain weight was observed in LLYHLZ (13.3%). Moreover, the reduction in seed setting rate and grain weight caused by elevated temperatures during the grain-filling stage was relatively lower compared to that observed during the booting stage.

### 3.2. Effects of Heat Stress on Appearance Quality

Significant differences (*p* < 0.05 to *p* < 0.001) were observed in chalkiness, chalky grain rate, grain length, grain width, and length/width ratio among varieties, temperature treatments, and their interactions, except that temperature treatments for length/width ratio and interactions regarding chalkiness and grain length (Table 1). In contrast to the control temperature, the higher temperature during the booting stage (HT1) led to significant increases in chalkiness (averaging 109.4%) and chalky grain rate (averaging 76.8%) across the four varieties, with the largest increases observed in YLY2646 for chalkiness and LLYHLZ for chalky grain rate. Additionally, HT1 resulted in a significant reduction in both grain length and grain width across the four varieties by 3.3%. However, significant decreases in the length/width ratio were observed in YLY2646, and a significant increase was observed in YLY2H under HT1. Conversely, there were no significant changes in the length/width ratio of QYSM and LLYHLZ when exposed to HT1.

Similar to the higher temperature experienced during booting, the elevated temperatures during grain filling also resulted in increased levels of chalkiness (averaging 196.9%) and chalky grain rate (averaging 140.4%) across all four varieties. Additionally, the higher temperature during grain filling significantly reduced the length/width ratio in YLY2646 exclusively. Furthermore, the higher temperature caused a notable decrease in grain length in YLY2646 (3.1%) and QYSM (1.4%), as well as a decrease in grain width in varieties except for LLYHLZ. Moreover, the increases in chalkiness and chalky grain rate induced by higher temperatures were more pronounced when the plants were exposed to higher temperatures during grain filling in comparison to elevated temperatures during booting; however, the opposite trend was observed regarding the reduction in grain length and width.

### 3.3. Effects of Heat Stress on Grain Nutritional Quality

Significant differences were observed in protein content, total starch content, amylose content, and amylopectin content among varieties and temperature treatments (Figure 2). When compared to the control treatment, higher temperatures during the booting stage (HT1) and grain-filling stage (HT2) resulted in a significant increase in protein content across all varieties. The increase in protein content was relatively higher under HT1 (with an average increase of 29.0% across all four varieties) compared to HT2 (with an average increase of 7.8% across all four varieties). Both HT1 and HT2 treatments led to a significant decrease in total starch content for all varieties, except for YLY2H under HT2. On average, HT1 and HT2 treatments resulted in a decrease of 6.4% and 1.5%, respectively, in the total starch content across the four varieties. Similarly, both HT1 and HT2 treatments caused a significant average decrease of 11.5% and 6.0% in amylose content for all varieties, respectively, except for LLYHLZ under HT2. HT1 treatment resulted in a significant decrease in amylopectin content across all varieties, with an average decrease of 5.2% in the four varieties. However, HT2 treatment did not have a significant effect on amylopectin content.

### 3.4. Effect of Heat Stress on Thermal Properties of Grain Starch

Various factors, including varieties, temperature treatments, and their interactions, had a significant impact on all thermal properties of grain starch, namely To, Tp, Tc, ΔH, R, and PHI (Table 2, *p* < 0.001 or *p* < 0.01 or *p* < 0.05). However, the effects of temperature treatments on R were not statistically significant. Compared to the control treatment, an elevated temperature during the booting stage (HT1) and grain filling (HT2) resulted in a significant increase in To for all varieties, except for To of YLY2646 under HT2. Additionally, during the booting stage, higher temperatures induced a significant increase in Tp and Tc for LLYHLZ and in Tp for YLY2646, whereas no significant changes were observed for the other varieties. In contrast to HT1, HT2 treatment caused an increase in Tp for YLY2646, QYSM, and YLY2H and an increase in Tc for YLY2646 and QYSM. Both HT1 and HT2 treatments significantly increased ΔH for YLY2646 and QYSM, while LLYHLZ showed an increase only under HT1; however, YLY2H did not exhibit any significant changes. Regarding R and PHI, there were no significant changes observed due to either higher temperature during booting HT1 or grain filling HT2 treatments for all varieties, except for R of YLY2646 under HT2 and PHI of QYSM and LLYHLZ under HT1.

### 3.5. Effects of Heat Stress on Pasting Characteristics of Grain Starch

As shown in Table 3, variety, temperature treatments, and their interactions had a significant effect on peak viscosity, trough viscosity, final viscosity, breakdown, and setback (*p* < 0.05, *p* < 0.01, or *p* < 0.001). When compared to the control temperature, higher temperature during the booting stage led to a significant increase in peak viscosity for YLY2H and LLYHLZ, whereas the differences were not significant in the other two rice varieties, namely YLY2646 and QYSM. On the other hand, higher temperatures during grain filling (HT2) resulted in a significant increase in peak viscosity across all varieties, with an average increase of 10.4% (Table 3). In terms of trough viscosity, high temperature during booting and grain filling led to a significant increase in all varieties except for HT1 for YLY2646. Additionally, the rise in trough viscosity was more pronounced under HT2 as compared to HT1. Final viscosity showed a significant increase in QYSM and YLY2H under HT1 in contrast to the control temperature. While YLY2646, QYSM, and YLY2H exhibited an increase in final viscosity under HT2. There were no significant changes observed in final viscosity for LLYHLZ under both HT1 and HT2 treatments. Regarding breakdown, HT1 treatment resulted in a significant decrease in QYSM and LLYHLZ, while HT2 treatment led to a significant increase in breakdown for all four rice varieties, except for LLYHLZ. In terms of setback, only YLY2646 showed a significant increase under HT1, while the other three varieties did not exhibit significant changes. And when exposed to HT2, QYSM, and YLY2H showed a significant reduction in the setback.

### 3.6. Principal Component and Correlation Analyses of Grain Yield and Quality Traits

The effects of high-temperature treatments during the booting and grain-filling stages on yield and quality traits of high-quality hybrid rice were studied by performing the principal component analysis (PCA) on three different temperature conditions (Figure 3). The results showed that PC1 and PC2 collectively explained 82.9% to 87.1% of the variations in all traits under different temperature conditions. Within the three temperature conditions, gelatinization temperature (To, Tp, and Tc), total starch content (TSC), amylopectin (AP), and breakdown (BD) were strongly positively correlated and clustered together on PC1 (Figure 4), indicating their close association. Conversely, ΔH, PHI, Fv, Tv, and SB showed strong negative correlations with the above-mentioned traits. GWBR and GW on PC2 exhibited significant negative correlations with GL and LWR. Additionally, the correlations between traits were influenced by the different temperature conditions. For instance, the yield trait SP was positively correlated with SSR and GWBR only under the HT2 condition. SSR, on the other hand, exhibited negative correlations with GWBR under the CK and HT1 conditions but showed a positive correlation with SP under the HT2 condition. Among the quality traits, CKY was unrelated to most indicators under the CK and HT1 conditions but positively correlated with gelatinization temperature and other indicators clustered together under the HT2 condition while negatively correlated with ΔH and PHI. Conversely, CGR showed positive correlations with gelatinization temperature and other indicators under the CK and HT1 conditions while negatively correlated with ΔH and PHI. However, it was mostly unrelated to other indicators under the HT2 condition. Furthermore, R was clustered with ΔH and PHI under the HT2 condition, indicating its association with other quality indicators, while showing little correlation with most quality traits under the CK and HT1 conditions. In contrast, PC exhibited fewer correlations with other indicators under the CK condition but demonstrated more associations with other quality indicators under high-temperature conditions.

Furthermore, differences were observed among the four varieties under different temperature conditions. For instance, LLYHLZ and YLY2H clustered together on PC1 under the CK condition, primarily exhibiting relatively high values for gelatinization temperature (To, Tp, Tc) and relatively low values for ΔH and PHI. In contrast, QYSM and YLY2646 showed the opposite pattern on PC1 compared to the former. Additionally, under the HT1 condition, LLYHLZ and YLY2H exhibited a separation on PC2 while maintaining their position on PC1. Specifically, YLY2H displayed a relatively higher yield. This separation became more pronounced under the HT2 condition.

## 4. Discussion

The booting and grain-filling stages are crucial growth phases in rice reproduction, greatly impacting grain development and being highly vulnerable to heat stress [20,23]. Our findings are consistent with previous studies that have demonstrated how heat stress during the booting stage primarily diminishes rice yield by reducing spikelets panicle^−1^, seed-set, and grain weight [30], while heat stress during the grain-filling stage primarily diminishes rice yield by reducing seed-set and grain weight [31]. Furthermore, our study reveals that a higher temperature during the booting stage has a more substantial impact on reducing rice yield compared to a higher temperature during grain filling. The primary reason for this disparity is that the number of spikelets is determined during the panicle initiation to the flowering stage [32]. Consequently, high temperatures during the booting stage can impede the transfer of dry matter from the stem sheath to the spikelets, leading to reduced differentiation of secondary rachis branches and spikelets in rice [33] and diminished pollen and stigma vigor, ultimately resulting in a decrease in spikelets per panicle and seed-set [22]. Higher temperature during the filling stage mainly hinders the development of endosperm cells and subsequently decreases the starch-accommodating capacity of the endosperm. Simultaneously, the shortened grain-filling stage or reduced grain-filling rate also affect seed-set and grain filling [34].

Compared to the control treatment, exposure to higher temperatures during the booting and grain-filling stages significantly increased the chalkiness and chalky grain rate of the grains. Additionally, the increase in chalkiness and chalky grain rate was higher during the grain-filling stage compared to the booting stage (Table 1). These findings align with previous studies that investigated the effects of higher temperatures during grain-filling or booting stages on chalkiness in Japonica rice [23,35,36]. The observed phenomenon can be attributed to heat stress during the grain-filling stage, which adversely affects grain filling and shortens its duration. As a result, the grains are not fully enriched, the starch structure becomes loose, and the gap between starch granules increases, leading to increased chalkiness [36,37]. The chalkiness induced by higher temperatures during booting may be attributed to unrecovered physiological and biochemical activities that affect photosynthetic capacity; this, in turn, results in a decrease in assimilate supply [38] and inhibits the remobilization of nonstructural carbohydrates into grains [23]. These factors adversely alter grain growth, ultimately causing and leading to the occurrence of chalkiness. Further research is needed to investigate the variations in the extent of rice chalkiness increase between higher temperatures during the grain-filling and booting stages. Moreover, higher temperatures during booting significantly reduced grain length and grain width across all varieties in contrast to the control treatment. On the other hand, higher temperatures during grain filling only affected one to two varieties in terms of grain length or width (Table 1). These results highlight the more pronounced impact of high-temperature stress during the booting stage on rice grain morphology. In addition to reducing assimilate accumulation and transport, high-temperature stress also hampers the development of the lemma and palea during the booting stage, resulting in a decrease in grain length and width [39]. After the grain-filling stage, the development of the lemma and palea is completed, and the grain growth is influenced only by assimilate transport and accumulation [40].

Higher temperatures have a noticeable impact on the nutritional composition of high-quality hybrid rice. In comparison to the control treatment, both the booting and grain-filling stages experience a significant rise in grain protein content under high-temperature stress. Notably, the increase in protein content during the booting stage surpasses that observed during the grain-filling stage. Prior studies have indicated that high temperatures during the booting stage can modify hormone levels in rice plants, resulting in an indirect inhibition of grain filling and starch formation while also increasing grain nitrogen concentration, thus promoting protein accumulation [23,41]. Conversely, under high-temperature stress during the grain-filling stage, increased activity of glutamate synthase (GOGAT) and transaminases facilitates nitrogen metabolism in the grains, promoting protein synthesis and accumulation [39,42]. Furthermore, high temperatures during the booting stage significantly reduced total soluble carbohydrates (TSC), amylose content (AM), and amylopectin content (AP) for all varieties compared to the control treatment, while high temperatures during the grain-filling stage significantly reduced TSC and AM for most varieties, although to a lesser extent than during the booting stage. Previous studies have indicated that high temperatures during the booting stage mainly reduce the size of the grain sink, thereby inhibiting carbohydrate accumulation in the grains [43,44] and significantly decreasing starch content in the grains [23]. Conversely, high temperatures during the grain-filling stage reduce starch content by inhibiting the expression of starch synthase I and branching enzyme genes and inducing the expression of α-amylase, thereby promoting starch consumption [36,37]. Moreover, previous studies have also found that high temperatures during the grain-filling stage do not significantly alter the amylopectin content in the grains [45], which is consistent with our research findings. Yoshida (1981) [46] reported that heat stress during the early reproductive stage has a more severe impact on rice yield formation compared to an equivalent treatment during the later reproductive stage. Similarly, our study found that the impact of high temperatures during the booting stage on grain protein and starch content in high-quality hybrid rice was greater than that during the grain-filling stage. This may be due to the direct limitation of the grain sink size by high-temperature stress during the booting stage, which profoundly affects subsequent assimilate transport and synthesis [21,43], while the damage to the grain sink capacity is relatively small after short-term high-temperature stress during the grain-filling stage, allowing for the compensation of assimilate losses during the stress period [47].

Overall, compared to the control treatment, higher temperatures during the booting stage significantly increased To and Tp in most varieties. Additionally, under the higher temperature during grain filling, To, Tp, and Tc significantly increased in all varieties compared to the control. Previous studies have found that the increase in gelatinization temperature caused by high-temperature stress during the grain-filling stage leads to a higher cooking temperature and longer cooking time for rice, resulting in a decline in rice quality [48,49]. This may be due to changes in the starch structure in rice grains grown under high temperatures, such as an increase in the proportion of large starch granules and average starch particle size, leading to an increase in gelatinization temperature [50,51]. Furthermore, previous studies have reported a significant negative correlation between amylose content and gelatinization temperature [52], and our results also found a significant negative correlation between amylose content and gelatinization temperature at both normal temperatures and higher temperatures during the booting. In addition, compared to the control temperature, higher temperature during the booting and grain filling significantly increased ΔH in most varieties. Previous research has suggested that a decrease in amylose content under high-temperature stress leads to an increase in the enthalpy of gelatinization ΔH [20], which is consistent with our research findings.

Via the examination of grain starch pasting characteristics, it was found that higher temperatures during the grain-filling stage significantly increased the peak, trough, and final viscosity in most varieties when compared to the control temperature. Conversely, higher temperatures during the booting stage only led to increases in specific varieties or caused smaller increments. These findings indicate that exposure to high temperatures during grain filling may result in rice grains exhibiting greater viscosity and higher viscosity levels during the gelatinization process. Previous studies have also revealed that high temperatures during the grain-filling stage led to heightened viscosity in semi-waxy japonica rice. These studies propose that this effect may be attributed to a reduction in amylose content and short-chain amylopectin, thereby resulting in increased gelatinization viscosity [19]. On the other hand, compared to the control temperature, higher temperature during grain filling significantly increased breakdown in most varieties, while under higher temperature during booting, certain varieties exhibited a noteworthy decrease in breakdown. Earlier research has proposed that changes in breakdown could be linked to starch content and composition [53], but the precise reasons for breakdown alterations under high-temperature stress remain uncertain and necessitate further investigation. Conversely, consistently high setback values were observed in all varieties subjected to higher temperatures during booting, while the control treatment generally yielded the lowest setback. Prior studies have indicated that the increase in the setback of rice starch under high-temperature conditions may be attributed to elevated protein content [54,55]. Overall, exposure to high-temperature conditions influences the viscosity characteristics of rice grains.

## 5. Conclusions

Heat stress during both the booting and grain-filling stages negatively affects the yield and quality of high-quality hybrid rice. The study revealed that elevated temperatures during these stages negatively impact the seed-setting rate and grain weight of the rice. Additionally, higher temperatures during the booting stage significantly reduce the number of spikelets per panicle. Furthermore, high temperatures contribute to a deterioration in rice quality. Specifically, during the grain-filling stage, the elevated temperatures result in increased chalkiness, higher gelatinization temperature, elevated protein content, decreased starch content, and heightened viscosity in the grains. Similarly, the booting stage under higher temperatures exhibits these changes, with a more pronounced impact on protein and starch content. Furthermore, the impact of high temperatures during the booting and grain-filling stages on rice grain quality also demonstrates variations among different rice varieties. The findings of this study emphasize the importance of considering the effects of high-temperature stress at different growth stages on both yield and quality when breeding heat-resistant rice. Moreover, further investigation is necessary to elucidate the mechanisms underlying the variations in rice quality characteristics caused by high temperatures during different stages.

## Figures and Tables

**Figure 1 foods-12-04093-f001:**
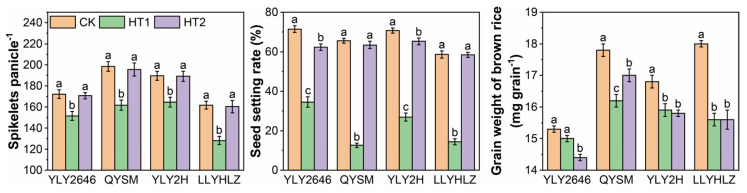
Effects of heat stress at booting (HT1) and grain-filling (HT2) stage on various yield parameters. Different alphabets assigned within each variety represent significant differences (*p* < 0.05) between the temperature treatments.

**Figure 2 foods-12-04093-f002:**
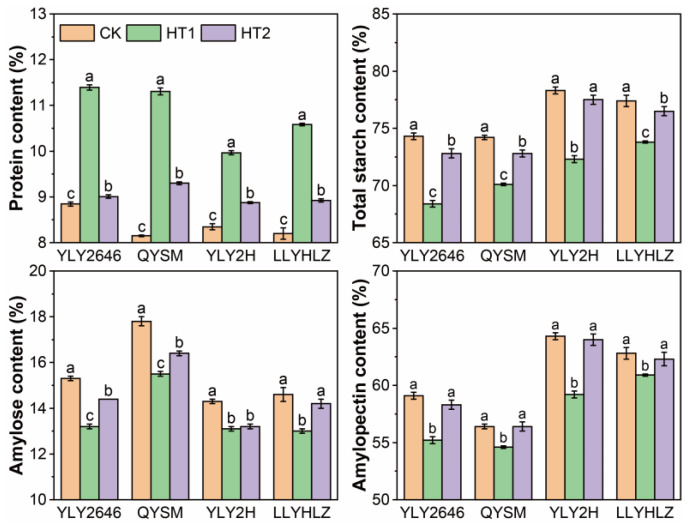
Effects of heat stress at booting (HT1) and grain-filling (HT2) stage on grain nutritional quality. Different alphabets assigned within each variety represent significant differences (*p* < 0.05) between the temperature treatments.

**Figure 3 foods-12-04093-f003:**
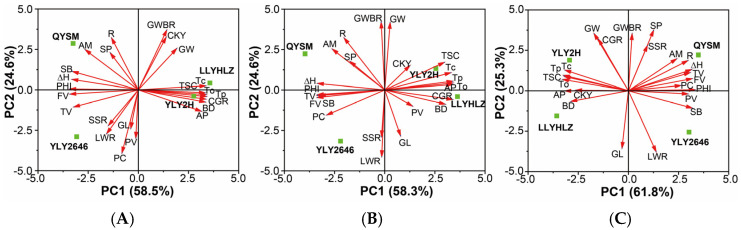
Principal component analysis showing biplot for analyzed grain yield and quality traits with varieties under control (**A**), the higher temperature during booting (**B**), and higher temperature during grain filling (**C**). Abbreviations: Spikelets panicle^−1^, SP; Seed setting rate, SSR; Grain weight of brown rice, GWBR; Chalkiness, CKY; Chalky grain rate, CGR; Length/width ratio, LWR; Grain length, GL; Grain width, GW; Protein content, PC; Total starch content, TSC; Amylose content, AM; Amylopectin content, AP.

**Figure 4 foods-12-04093-f004:**
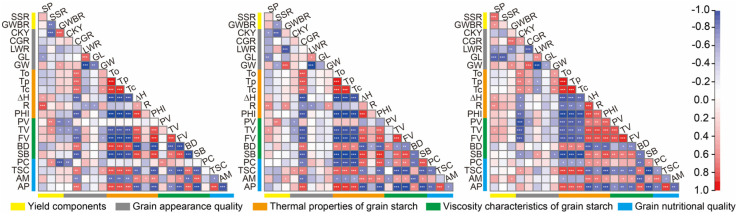
Correlation matrix showing Pearson’s correlation among traits in the high-quality hybrid rice under control (CK), higher temperature during booting (HT1), and grain-filling stage (HT2), where *, **, and *** indicate significant correlation at *p* < 0.05, < 0.01, and < 0.001, respectively. The scale bar on the right indicates the intensity of the correlation from 1 (highest positive) to −1 (highest negative). Abbreviations: chalkiness, CKY; chalky grain rate, CGR; length/width ratio, LWR; grain length, GL; grain width, GW; protein content, PC; total starch content, TSC; amylose content, AM; amylopectin content, AP.

**Table 1 foods-12-04093-t001:** Effect of control (CK) and higher temperatures during the booting stage (HT1) or grain-filling stage (HT2) on the grain appearance quality.

Variety	Temperature Treatments	Chalkiness (%)	Chalky Grain Rate (%)	Grain Length (mm)	Grain Width (mm)	Length/Width Ratio
YLY2646	CK	9.77 ± 2.29 c	24.15 ± 3.53 c	6.72 ± 0.02 a	1.81 ± 0.00 a	3.71 ± 0.01 a
	HT1	22.56 ± 0.61 b	44.16 ± 2.64 b	6.40 ± 0.02 c	1.78 ± 0.01 b	3.59 ± 0.02 b
	HT2	31.64 ± 0.36 a	51.51 ± 3.17 a	6.51 ± 0.03 b	1.79 ± 0.01 b	3.63 ± 0.03 b
QYSM	CK	14.20 ± 0.73 c	22.99 ± 1.67 c	6.49 ± 0.04 a	2.03 ± 0.00 a	3.19 ± 0.02 a
	HT1	27.88 ± 3.06 b	39.31 ± 4.99 b	6.32 ± 0.02 b	2.01 ± 0.01 b	3.15 ± 0.02 a
	HT2	36.14 ± 1.22 a	68.04 ± 1.08 a	6.40 ± 0.01 b	2.01 ± 0.01 b	3.18 ± 0.02 a
YLY2H	CK	11.63 ± 0.86 c	29.49 ± 3.25 c	6.43 ± 0.05 a	2.13 ± 0.00 a	3.02 ± 0.03 b
	HT1	24.24 ± 1.72 b	50.41 ± 2.08 b	6.28 ± 0.04 b	2.00 ± 0.01 c	3.14 ± 0.02 a
	HT2	37.57 ± 1.06 a	67.01 ± 1.74 a	6.33 ± 0.03 ab	2.09 ± 0.01 b	3.02 ± 0.02 b
LLYHLZ	CK	15.65 ± 1.24 c	29.24 ± 1.98 c	6.68 ± 0.11 a	1.99 ± 0.02 a	3.36 ± 0.09 a
	HT1	31.73 ± 1.57 b	53.44 ± 1.83 b	6.45 ± 0.03 b	1.90 ± 0.01 b	3.40 ± 0.02 a
	HT2	45.02 ± 1.61 a	66.00 ± 1.07 a	6.58 ± 0.03 ab	1.97 ± 0.02 a	3.33 ± 0.03 a
Variety (V)		***	***	***	***	***
Temperature treatment (T)		***	***	***	ns	***
V × T		ns	*	ns	*	***

Data shown were mean ± standard error. Within a variety, means followed by the same alphabet were not significantly different at *p* = 0.05. * *p* < 0.05; *** *p* < 0.001; ns, not significant (*p* > 0.05).

**Table 2 foods-12-04093-t002:** Effect of control (CK) and higher temperatures during the booting stage (HT1) or grain-filling stage (HT2) on the thermal properties of grain starch.

Variety	Temperature Treatments	To (°C)	Tp (°C)	Tc (°C)	∆H (J/g)	R	PHI
YLY2646	CK	68.73 ± 0.05 b	73.70 ± 0.04 b	78.13 ± 0.09 b	4.89 ± 0.01 b	9.36 ± 0.06 b	0.99 ± 0.01 ab
	HT1	69.30 ± 0.03 a	74.37 ± 0.07 a	78.64 ± 0.08 ab	5.18 ± 0.03 a	9.34 ± 0.08 b	1.02 ± 0.02 a
	HT2	68.74 ± 0.11 b	73.99 ± 0.17 a	79.36 ± 0.21 a	5.13 ± 0.01 a	10.61 ± 0.10 a	0.98 ± 0.01 b
QYSM	CK	67.27 ± 0.16 c	72.52 ± 0.10 b	78.93 ± 0.13 b	5.32 ± 0.04 c	11.66 ± 0.05 a	1.01 ± 0.01 b
	HT1	67.74 ± 0.24 b	73.03 ± 0.24 b	78.94 ± 0.59 b	5.95 ± 0.04 b	11.20 ± 0.42 a	1.13 ± 0.01 a
	HT2	69.81 ± 0.18 a	75.94 ± 0.23 a	81.46 ± 0.33 a	6.14 ± 0.05 a	11.65 ± 0.19 a	1.00 ± 0.02 b
YLY2H	CK	77.56 ± 0.23 c	83.75 ± 0.27 b	88.02 ± 0.27 a	3.97 ± 0.01 a	10.46 ± 0.18 a	0.64 ± 0.00 a
	HT1	78.14 ± 0.07 b	84.08 ± 0.18 b	88.39 ± 0.29 a	4.03 ± 0.02 a	10.25 ± 0.22 a	0.68 ± 0.02 a
	HT2	78.83 ± 0.18 a	85.07 ± 0.24 a	88.82 ± 0.32 a	4.02 ± 0.02 a	9.99 ± 0.15 a	0.64 ± 0.01 a
LLYHLZ	CK	77.83 ± 0.01 c	83.71 ± 0.05 b	87.08 ± 0.09 b	3.40 ± 0.10 b	9.25 ± 0.07 a	0.58 ± 0.02 b
	HT1	79.07 ± 0.13 a	84.74 ± 0.20 a	87.97 ± 0.34 a	4.08 ± 0.07 a	8.90 ± 0.21 a	0.72 ± 0.00 a
	HT2	78.30 ± 0.06 b	84.20 ± 0.03 b	87.34 ± 0.10 ab	3.47 ± 0.05 b	9.04 ± 0.04 a	0.59 ± 0.01 b
Variety (V)		***	***	***	***	***	***
Temperature treatment (T)		***	***	***	***	ns	***
V × T		***	***	*	***	***	**

The data shown were mean ± standard error. Within a variety, means followed by the same alphabet were not significantly different at *p* = 0.05. * *p* < 0.05; ** *p* < 0.01; *** *p* < 0.001; ns, not significant (*p* > 0.05).

**Table 3 foods-12-04093-t003:** Effect of control (CK) and higher temperatures during the booting stage (HT1) or grain-filling stage (HT2) on viscosity characteristics of grain starch.

Variety	Temperature Treatments	Peak Viscosity (cP)	Trough Viscosity (cP)	Final Viscosity (cP)	Breakdown (cP)	Setback (cP)
YLY2646	CK	3212.3 ± 21.9 b	2103.7 ± 22.7 b	2895.0 ± 6.4 b	1108.7 ± 38.8 b	786.5 ± 3.2 b
	HT1	3222.5 ± 1.4 b	2108.5 ± 3.2 b	2946.7 ± 16.1 ab	1114.0 ± 4.6 b	843.0 ± 8.4 a
	HT2	3603.0 ± 36.5 a	2236.3 ± 31.9 a	3027.3 ± 43.3 a	1366.7 ± 5.9 a	791.0 ± 13.1 b
QYSM	CK	2983.7 ± 77.1 b	2003.7 ± 10.4 c	2863.3 ± 21.4 c	1003.7 ± 6.5 b	859.7 ± 13.9 a
	HT1	3007.3 ± 10.2 b	2115.7 ± 27.3 b	2997.7 ± 10.2 b	868.0 ± 99.8 c	882.0 ± 20.1 a
	HT2	3497.7 ± 13.9 a	2307.0 ± 20.2 a	3076.0 ± 18.4 a	1190.7 ± 9.2 a	769.0 ± 3.1 b
YLY2H	CK	3255.3 ± 5.4 b	1813.0 ± 33.8 c	2385.7 ± 20.5 c	1357.0 ± 9.6 b	646.0 ± 2.6 a
	HT1	3349.3 ± 4.2 a	1898.3 ± 14.9 b	2544.3 ± 17.4 b	1337.3 ± 16.2 b	650.7 ± 4.2 a
	HT2	3372.3 ± 15.0 a	2012.0 ± 20.0 a	2662.7 ± 21.0 a	1559.3 ± 48.8 a	572.7 ± 13.3 b
LLYHLZ	CK	2953.0 ± 20.8 c	1646.7 ± 15.2 b	2310.7 ± 16.2 a	1436.3 ± 18.4 a	645.3 ± 8.3 ab
	HT1	3083.0 ± 3.5 b	1729.7 ± 26.4 a	2364.0 ± 28.1 a	1223.3 ± 17.7 b	664.0 ± 3.1 a
	HT2	3208.7 ± 31.4 a	1741.7 ± 20.3 a	2375.0 ± 27.6 a	1467.0 ± 27.5 a	622.3 ± 10.2 b
Variety (V)		***	***	***	***	***
Temperature treatment (T)		***	***	***	**	***
V × T		***	***	***	***	*

The data shown were mean ± standard error. Within a variety, means followed by the same alphabet were not significantly different at *p* = 0.05. * *p* < 0.05; ** *p* < 0.01; *** *p* < 0.001.

## Data Availability

Data is contained within the article.

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
