# Peer review of "Comparative Effects of Heat Stress at Booting and Grain-Filling Stage on Yield and Grain Quality of High-Quality Hybrid Rice"

_foods, 2023, doi:10.3390/foods12224093_

Round 1

Reviewer 1 Report

Comments and Suggestions for Authors

Dear authors,
After reading and analyzing your manuscript (Comparative effects of heat stress at booting and grain filling stage on yield and grain quality of high-quality hybrid rice), In general, this work is interesting, besides I think it has scientific quality. I give you the following remarks:

1.     The title does not accurately reflect the scientific study, please modify it and simplify.

2.     The abstract is perfect.

3.     The introduction is perfect even if it is a little long.

4.     In part 2.1. Plant materials and experimental design, if you still have images of seeds grown especially during heat stress, please add them.

5.     The two parts 2.2 Sampling and yield observations and 2.3. Appearance quality are perfect.

6.     Is there an international or Chinese standard for measuring thermal properties, thank you for the one cited in part 2.4. Thermal properties analysis.

7.     the figures are a little blurry.

8.     The titles of the figures are long, please delete the last part.

9.     In part 3.3. Effects of heat stress on grain nutritional quality, please check the numbering of figure 1 by figure 2.

10. In part 4. Discussion please explain this phenomenon to us "Previous studies have suggested that during the booting stage, high temperatures inhibit starch accumulation, favoring protein accumulation.

11. Part 4. General discussion is well written and explains the results well in comparison with previous studies on the effect of heat stress on rice yield and grain quality.

12. The conclusion and references are perfect.

Best regards

Reviewer 2 Report

Comments and Suggestions for Authors

At the methodology please add information regarding chemistry composition of soil profile and soil pH. That information are important to have a general view regarding heat stress at booting and grain filling stage.

Reviewer 3 Report

Comments and Suggestions for Authors

The manuscript seems interesting and well-written. However, it needs a minor revision.

Line 14: as well as the different effects of the booting……………

Introduction

Line 78: conducted by Zhen et al. [23] has revealed………

Materials and Methods

Line 182: the method by Lan et al. [26] with slight……..

Results:

Line 289: higher temperature during the booting stage led to a significant increase in peak viscosity?? Why?

Conclusions:

Line 479: Furthermore, high temperatures contribute to a deterioration………

The manuscript seems well written.

Overall Comments:

  1. All the references should be uniform across the manuscript.
  2. English language must be concise.
  3. The name of all equipment should be mentioned along with their company names and model numbers.

Comments on the Quality of English Language

English needs to be revised for grammatical errors.

Reviewer 4 Report

Comments and Suggestions for Authors

Manuscript recieved for review investigates different rice varieties, which were exposed to control temperature and heat stress during booting and grain filling stages, for different quality parameters. 

Title is appropriate.

Introduction section is too large, it needs revision.

Material and methods corresopnd to investigated material and conducted testing. 

Results are claerly presented. Applied statistical methods underline differences among different varieties. 

Discussion needs some corrections, noted in manuscript pdf file. 

Conclusion section, also needs some corrections. 

References are numerous, adequate and latest.

Comments on the Quality of English Language

Some typing errors
